# The Clinical Implication of Serogroup Distribution and Drug Resistance of Non-Typhoidal *Salmonella* in Children: A Single Center Study in Southern Taiwan during 2004–2019

**DOI:** 10.3390/children9091403

**Published:** 2022-09-16

**Authors:** Meng-Chien Lee, Zon-Min Lee, Yi-Chun Yeh, Hong-Ren Yu, Kuang-Che Kuo

**Affiliations:** 1Department of Emergency Medicine, An Nan Hospital, China Medical University, Tainan 70965, Taiwan; 2Department of Pharmacy, Kaohsiung Chang Gung Memorial Hospital, Kaohsiung 83301, Taiwan; 3Department of Psychiatry, Kaohsiung Medical University Hospital, Kaohsiung 80756, Taiwan; 4Department of Psychiatry, School of Medicine, College of Medicine, Kaohsiung Medical University, Kaohsiung 80756, Taiwan; 5Department of Pediatrics, Kaohsiung Chang Gung Memorial Hospital, Kaohsiung 83301, Taiwan; 6Department of Pediatrics, Kaohsiung Chang Gung Memorial Hospital and Chang Gung University College of Medicine, Kaohsiung 83301, Taiwan

**Keywords:** non-typhoidal *Salmonella*, pediatric, emerging serotypes, antimicrobial susceptibility

## Abstract

Background: A regional antibiotic susceptibility database of certain pathogens is crucial for first-line physicians in terms of providing clinical judgement and appropriate selection of antimicrobial agents. The aim of this study is to update the epidemiological data of *Salmonella* serogroups and drug resistance in pediatric patients. Methods: This is a single-center retrospective study enrolling patients aged from 0 to 18 years who were hospitalized with cultured proven non-typhoidal *Salmonella* (NTS) infection from 2004 to 2019. The isolates were collected and the demographic data, serogroups of *Salmonella* and antimicrobial susceptibilities were further analyzed. Results: A total of 1583 isolates of NTS were collected. Serogroup C2 was prone to cause invasive non-typhoidal salmonellosis (iNTS), especially bacteremia. Patients aged < 2 years were associated with serogroups B and C2 infection, while those aged ≥ 2 years were associated with serogroups D and E infection. The prevalence of serogroup B declined with simultaneous increase in prevalence of serogroups D and E. Serogroups B and E were associated with ceftriaxone resistance, while Serogroup D was less drug-resistant than the others. The prevalence of ceftriaxone-resistant *Salmonella* had not increased, although more ciprofloxacin-resistant isolates were found in iNTS infection. Conclusions: Age < 2 years is a risk factor of iNTS for children, and the distribution of serogroup changes should be closely monitored. Ceftriaxone is still the drug of choice for treating pediatric iNTS infection, and although no increase was observed in the prevalence of ceftriaxone-resistant strains in this study, continuing surveillance of such cases is warranted.

## 1. Introduction

Non-typhoidal *Salmonella* is one of the major pathogens of food-borne infection worldwide. It is estimated 94 million cases of gastroenteritis are caused by non-typhoidal *Salmonella* (NTS) worldwide annually [1,2,3]. Bacterial enteric infections are an important cause of illness in children. A study from the United States enrolling children under 5 years of age with laboratory-confirmed bacterial enteric illness reported that NTS was the most commonly isolated (42%) bacterial enteric pathogen, followed by *Campylobacter* (28%), *Escherichia coli* 0157, *Shigella* and *Yersinia enterocolitica* [4]. In Taiwan, NTS remains the leading pathogen of childhood bacterial enterocolitis requiring hospitalization [5]. NTS not only causes enteric infection in children, but also infects normally sterile sites, such as the blood (bacteremia), the synovial fluid (arthritis) or the cerebrospinal fluid (meningitis), which is defined as invasive non-typhoidal salmonellosis (iNTS).

The global increase of antimicrobial resistance in NTS is a warning to clinicians. NTS has now been reported to be commonly resistant to antimicrobials such as ampicillin, trimethoprim-sulfamethoxazole (TPM/SMX), and chloramphenicol [6]. Additionally, in recent years, it has been reported to be resistant to commonly used first-line antimicrobials such as ciprofloxacin and cefotaxime/ceftriaxone, which represents a threat to the efficacy of NTS infection treatments [7,8].

A regional antibiotic susceptibility database of certain pathogens is crucial to first-line physicians for the clinical assessment and selection of appropriate antimicrobial agents, as is knowledge of local epidemiological and antimicrobial resistance trends of NTS serogroups. In this study, the susceptibility of NTS to different categories of antibiotics has been reviewed. Further, the trend of susceptibility to commonly used antimicrobials, the prevalence and trends of different serogroups and the drug resistance of different serogroups of NTS have also been further analyzed.

## 2. Materials and Methods

This retrospective study was conducted at Kaohsiung Chang Gung Memorial Hospital (KCGMH), a tertiary referral hospital in southern Taiwan, from October 2004 to March 2019, and enrolled children aged 0–18 years, hospitalized due to culture-proven NTS infection. Each isolate was collected from a single patient with unique sampling site and date. Isolates from each patient but at different sampling sites or dates were all kept for data analysis. Patients with underlying diseases including cancer, leukemia, diffuse disease of connective tissue, cerebral palsy, biliary atresia and Hirschsprung disease were excluded.

Clinical information was retrieved by retrospectively reviewing medical charts including demographic details, susceptibility to antibiotics and NTS serogroups. Bacterial identification and antibiotics susceptibility testing were conducted by the microbial laboratory of KCGMH. All isolates were cultured and identified according to standard methods, with no major changes over time in the policy for the identification of Salmonella. Serotyping was done by Wellcolex color *Salmonella* test (Murex, Dartford, UK), then confirmed by slide agglutination test with the use of O antiserum to detect O antigen (Bacto, Liverpool, NSW, Australia). NTS isolates were then classified into serogroups including serogroup B, C1, C2, D, E and other serogroups. The antimicrobial susceptibility of *Salmonella* isolates was examined by standard disc-diffusion method. Resistance to a specific antimicrobial was based on reference zone diameter interpretive standards based on Clinical and Laboratory Standards Institute (CLSI) guidelines from 2004 and updated every year [9]. In this study, isolates considered as intermediate or resistant strains according to the criteria of the CLSI guidelines were regarded as non-susceptible or resistant to certain antibiotics. The antimicrobial agents examined were ampicillin, chloramphenicol, ceftriaxone, ciprofloxacin and trimethoprim-sulfamethoxazole. For comparing the antimicrobial susceptibility and epidemiology of NTS serogroup, the study period was divided into two for analysis (period 1: 2004–2011 and period 2: 2012–2019). Ethical approval for this study was granted by the Institutional Review Board of Chang Gung Memorial Hospital (No. 201801208B0C501), Taiwan.

Statistical analyses were performed using SPSS software version 21.0 (SPSS Inc., Chicago, IL, USA). Chi-square test and Fisher’s exact test were used to compare the categorical variables. Continuous variables were compared using Mann-Whitney U test and the results were presented as median. The linear-by-linear association test was used for analysis of the trend. A *p* value of <0.05 was considered as statistical significance.

## 3. Results

A total of 1583 isolates of *Salmonella enterica* were collected. Among them, 887 (56.0%) were from males and the mean age was 2.1 years, while 1500 were cultured from stool samples, 66 from blood, 9 from urine, 2 from ascites, 2 from cerebrospinal fluid, 2 from synovial fluid and 2 from pus. The most common serogroup of *Salmonella enterica* was serogroup B (*n* = 688, 43.4%), followed by serogroup D (*n* = 481, 30.4%), serogroup C2 (*n* = 207, 13.1%), serogroup C1 (*n* = 139, 8.8%), and serogroup E (*n* = 68, 4.3%). Within 66 isolates from blood, 25 (37.9%) were serogroup B, followed by serogroup C2 (*n* = 19, 28.8%), serogroup D (*n* = 15, 22.7%), and serogroup C1 (*n* = 7, 10.6%). Of 9 isolates from urine, 5 were serogroup B, 2 were serogroup D, 1 was serogroup C1 and 1 was serogroup E. Both the isolates cultured from cerebrospinal fluid were serogroup B.

There was no statistical correlation between sex in the different serogroups. By comparing the age groups < 2 years to age ≥ 2 years, the former was more associated with serogroups B (*p* = 0.002, 46.7% vs. 38.9%) and C2 (*p* = 0.050, 14.5% vs. 11.1%), while the latter was more associated with serogroups D (*p* = 0.015, 33.7% vs. 28.0%) and E (*p* = 0.000, 7.1% vs. 2.3%).

The prevalence of serogroups in each year is analyzed and shown in Table 1.

The prevalence of serogroups B, D and E showed significant differences between years (*p* = 0.000, *p* = 0.001 and *p* = 0.000 respectively). By post hoc comparison, the prevalence of serogroup B demonstrated a decreasing trend (*p* = 0.000), while that of serogroups D and E demonstrated an increasing trend (*p* = 0.003 and *p* = 0.000 respectively). The prevalence of each serogroup was further analyzed by splitting the period of study into two (period 1: 2004~2011; period 2: 2012~2019). C1 and C2 showed no differences in prevalence between the two periods. Serogroup B showed lower prevalence in period 2 than in period 1, with statistical significance (*p* = 0.000, period 1: 47.2%; period 2: 38.1%). Serogroups D and E showed higher prevalence in period 2 than in period 1 (*p* = 0.009 and *p* = 0.000 respectively; serogroup D: 27.8% in period 1 vs. 34.0% in period 2; serogroup E: 2.2% in period 1 vs. 7.3% in period 2). 

The susceptibility of different serogroups of salmonella to five categories of antimicrobials was examined. Overall, the non-susceptible rate to ampicillin was 42.9%, followed by 30.1% to chloramphenicol, 23.5% to TPM/SMX, 5.9% to ceftriaxone and 2.0% to ciprofloxacin.

The percentages of non-susceptible rate of salmonella to different antimicrobials in each year are shown in Figure 1.

The differences of the prevalence of non-susceptible strains to each antimicrobial were further compared by dividing into two periods. The percentages of non-susceptible isolates to ampicillin showed a significant increase in period 2 compared to period 1 (*p* = 0.000, 38.8% in period 1 vs. 48.6% in period 2). There were no significant differences in the prevalence of non-susceptible strains to ciprofloxacin, ceftriaxone and TPM/SMX between 2 periods.

The association between each serogroup and non-susceptibilities to different antimicrobials are shown in Table 2. It may be observed that serogroup B was significantly associated with non-susceptibility to ampicillin, chloramphenicol, ceftriaxone and TPM/SMX compared to the other serogroups.

The prevalence of the non-susceptible isolates in different serogroups to antimicrobials in each year are shown in Figure 2. 

The differences of prevalence were compared by dividing into two periods (period 1: 2004–2011; period 2: 2012–2019). For serogroup B, there was a decrease in resistance rate to chloramphenicol and trimethoprim-sulfamethoxazole (chloramphenicol: *p* = 0.000, 59.9% in period 1 vs. 37.2% in period 2; trimethoprim-sulfamethoxazole: *p* = 0.001, 34.4% in period 1 vs. 22.0% in period 2). For serogroup C1 and C2, there were no significant differences in the prevalence of non-susceptible strains to each antimicrobial between the two periods. 

As with serogroup D, there was a significant increase in prevalence of non-susceptible strains to ampicillin and trimethoprim-sulfamethoxazole (ampicillin: *p* = 0.000, 8.9% in period 1 vs. 34.5% in period 2; trimethoprim-sulfamethoxazole: *p* = 0.011, 5.8% in period 1 vs. 12.6% in period 2). For serogroup E, a significant increase in prevalence of non-susceptible strains to ampicillin, chloramphenicol and trimethoprim-sulfamethoxazole was found (ampicillin: *p* = 0.000, 25.0% in period 1 vs. 75.0% in period 2; chloramphenicol: *p* = 0.000, 10.0% in period 1 vs. 75.0% in period 2; trimethoprim-sulfamethoxazole: *p* = 0.000, 15.0% in period 1 vs. 75.0% in period 2). 

The isolates from patients with iNTS were further compared to the non-iNTS group. There were no significant differences in sex between the two groups. Patients <2 years were associated with more iNTS compared to age ≥ 2 years (*p* = 0.000, 7.8% vs. 0%). Except for serogroup C2, there was no difference in serogroup distribution between iNTS and non-iNTS groups. Isolates of iNTS were more associated with serogroup C2 compared to non-iNTS isolates with statistical significance (*p* = 0.001, 27.8% vs. 12.4%). Also, serogroup C2 was associated with more bacteremia infection than other serogroups (*p* = 0.001, 9.2% vs. 3.4%). As with antimicrobial susceptibilities, isolates of iNTS were less-resistant to ampicillin compared to non-iNTS isolates (*p* = 0.020, 29.2% vs. 43.5%). On the contrary, isolates of iNTS were associated with more resistance to ciprofloxacin compared to non-iNTS isolates (*p* = 0.000, 15.3% vs. 1.4%). The distribution of serogroups and antibiotic non-susceptibility of both iNTS and non-iNTS isolates is further summarized in Table 3.

## 4. Discussion

This study provides a database of the antimicrobial susceptibility of NTS in pediatric patients in southern Taiwan and information regarding different serogroups and their antimicrobial susceptibilities with a relatively long period of time of surveillance. However, this is a retrospective, single-center study for clinical practice, and the results should be interpreted and used with care due to regional differences. The serotyping of salmonella was not further performed in this hospital, so epidemiologic details of the different serotypes in each serogroup were limited. 

In this study, serogroup B was the most common serogroup cultured from pediatric patients, accounting for 43.4% of cases. This is in line with results published elsewhere [10,11,12]. However, this result is quite different from that based on research performed in a medical center in northern Taiwan, where serogroup D was found to be the most common serogroup, accounting for 42.6% in children [13]. Serogroup D has also been reported to be the most prevalent serogroup among Turkish children [14]. From our research, patients aged < 2 years were associated with more serogroup B and C2 infection. On the other hand, patient aged ≥ 2 years were associated with more serogroup group D and E infection. *S. Typhimurium* (serogroup B) and *S. Newport* (serogroup C2) are the two most common serotypes in infants, as reported by Jones et al., which is consistent with our results [15].

The prevalence of serogroup B *Salmonella* declined, but serogroups D and E *Salmonella* demonstrated an increasing trend in our study. Except for the increase of serogroup E, our results of declining prevalence of serogroup B with concomitant increase in prevalence of serogroup B are consistent with those of many studies [11,16].

*S. Dublin* (serogroup D) and *S. Choleraesuis* (serogroup C1) have been reported to be more invasive than other serotypes [15]. In our research, the percentages of serogroups C1 and D within blood isolates were less than serogroups B and C2. Although serogroup B remained the most common serogroup within iNTS isolates, serogroup C2 accounted for a greater proportion within iNTS isolates than other serogroups, as reported by other studies [10,12]. Serogroup C2 was found to be more prone to cause bacteremia infection than other serogroups; however, this result is in contrast to research in northern Taiwan, where serogroup D was found to be more prone to cause bacteremia [17]. *S. Choleraesuis* (group C1) was found to be more likely to cause hospitalization than other serotypes [15]. Albeit without statistical significance, the prevalence of serogroup C1 seemed to be increased, from 7.3% in period 1 to 23.5% in period 2 within iNTS. For non-typhoid *Salmonella* meningitis, two of the cases in our study were infected by serogroup B. A study which included 24 *Salmonella* meningitis infants from Changhua (central Taiwan) reported that serogroup D (41.7%) was the most common pathogen causing meningitis followed by serogroup B (12.5%) [18]. *S. Typhimurium* (group B) is the most common pathogen causing *Salmonella* meningitis in South Africa followed by *S. Enteritidis* (group D) [19].

The antimicrobial resistance of the zoonotic pathogen *Salmonella* has been a global issue in recent decades. This resistance might be influenced by medical practice, veterinary medicine and the food industry. Conventional antimicrobials that have been widely used for decades such as ampicillin, chloramphenicol or trimethoprim-sulfamethoxazole showed high drug resistance within *Salmonella* serogroups, consistent with the findings of previous studies [10,20,21,22]. Compared with another study performed in southern Taiwan, the overall resistant rates of ampicillin, chloramphenicol or trimethoprim-sulfamethoxazole were similar; however, the non-susceptible rate of *Salmonella* to ampicillin increased in period 2 compared to period 1 in our study, which was not observed in the comparison study. The differences might partially have been caused by different regional clinical practices or increasing prevalence of serogroup E observed in our study, which is associated with ampicillin resistance. For more definitive reasons, continuing surveillance is required. Due to the relatively high resistance rate of *Salmonella* to conventional antibiotics, third-generation cephalosporins and fluoroquinolones have become the drug of choice to treat *Salmonella* presenting with serious disease [23]. However, over concern for the adverse effect on development of cartilage, clinical use of fluoroquinolones in children is limited [24]. Consequently, third-generation cephalosporins are the most commonly used antimicrobial to treat pediatric invasive non-typhoidal salmonellosis [25]. 

Ceftriaxone resistance of *Salmonella* is a public health issue, especially for children. Increasing ceftriaxone resistance was found to be spread by self-transferable beta-lactamase genes AmpC and blaCMY-2 [26]. Current research has mainly focused on the presence of blaCMY-2, which has been associated with resistance to extended spectrum cephalosporin [27]. The presence of this gene appears to reduce susceptibility to ceftriaxone [23]. Several *Salmonella* serotypes such as Typhimurium (group B), Agona (group B), and Newport (group C2) carry this gene [27]. Increasing ceftriaxone resistance in *Salmonella* has been reported in northern Taiwan where ceftriaxone resistance increased from <5% to >10% from 1999 to 2010 [28]. In the present study, the increase in resistance to ceftriaxone has not been observed, although the overall resistant rate to ceftriaxone seemed to be higher (5.9%) in our study compared to a study performed in another medical center in southern Taiwan but in a different city (1.1%) [11]. The ceftriaxone resistance was found to be more in serogroup B than other serogroups in our study, which is consistent with the report mentioned earlier [27]. In contrast, serogroup D showed less ceftriaxone resistance and less resistance to all the other antimicrobials in our study, which is different from many studies that have stated both serogroups B and D are more ceftriaxone-resistant than others [17,28]. 

Although fluoroquinolone carries possible risk for pediatric cartilage development, it is still an alternative for treatment, especially when clinicians are faced with the threat of ceftriaxone-resistant *Salmonella* infection. In our research, the resistance rate of ciprofloxacin was found to be 2%, and the rate seemed to not be increasing. The overall resistance rate to ciprofloxacin seemed to be higher within iNTS isolates compared to non-iNTS isolates. Serogroup E was found to be associated with more ciprofloxacin resistance than other serogroups. Meanwhile, the prevalence of serogroup E increased. The trend of resistance to fluoroquinolone should be further monitored, since increasing resistance to ciprofloxacin has been reported, especially in China [29]. 

## 5. Conclusions

Treating iNTS with ceftriaxone seems to be more promising compared to ciprofloxacin, which has a relatively higher resistance rate, although owing to increasing ceftriaxone-resistant *Salmonella* reported by many studies, continued surveillance and monitoring are still warranted. Serogroup B, which is associated with ceftriaxone resistance, declined in prevalence in our study. Serogroup C2, which is an emerging serogroup, was prone to cause bacteremia; however, no increase in prevalence or specific drug resistance was found in the current study. Serogroup D, which has been reported to be prone to carry ceftriaxone-resistant genes which cause invasive disease, seemed to be less drug-resistant and relatively not as invasive. The possible reasons for the differences require further investigation. Serogroup E demonstrated more ciprofloxacin resistance in our study, and both serogroups D and E revealed increased prevalence, indicating the need for further monitoring and surveillance in the future.

## Figures and Tables

**Figure 1 children-09-01403-f001:**
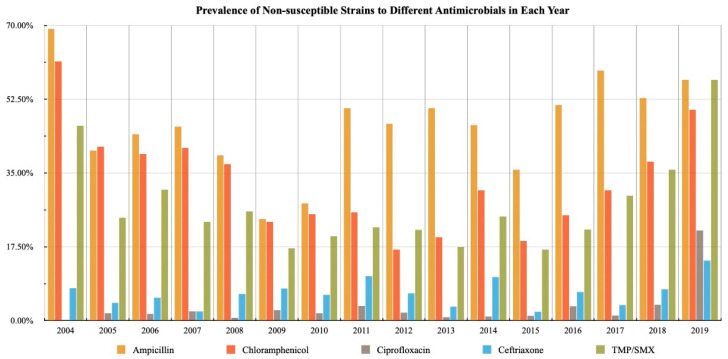
Prevalence of non-susceptible strains to different antimicrobials in each year.

**Figure 2 children-09-01403-f002:**
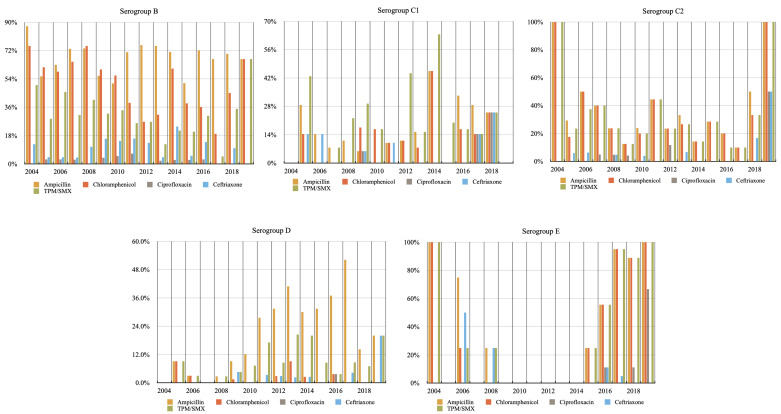
The prevalence of the non-susceptible isolates in different serogroups to antimicrobials in each year.

**Table 1 children-09-01403-t001:** The prevalence of different serogroups in each year.

Year Serogroup	2004	2005	2006	2007	2008	2009	2010	2011	2012	2013	2014	2015	2016	2017	2018	2019	Total	The Trend of Prevalence
Serogroup B	8 (61.5%)	70 (58.8%)	69 (53.5%)	74 (54.0%)	64 (44.8%)	50 (31.6%)	41 (35.7%)	62 (54.9%)	45 (42.1%)	48 (39.7%)	38 (39.2%)	39 (41.1%)	36 (40.9%)	21 (25.9%)	20 (37.7%)	3 (21.4%)	688	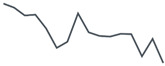
Serogroup C1	0 (0%)	7 (5.9%)	7 (5.4%)	13 (9.5%)	18 (12.6%)	17 (10.8%)	6 (5.2%)	10 (8.8%)	9 (8.4%)	13 (10.7%)	11 (11.3%)	10 (10.5%)	6 (6.8%)	7 (8.6%)	4 (7.5%)	1 (7.1%)	139	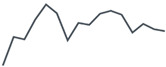
Serogroup C2	1 (7.7%)	17 (14.3%)	16 (12.4%)	20 (14.6%)	21 (14.7%)	24 (15.2%)	25 (21.7%)	9 (8.0%)	17 (15.9%)	15 (12.4%)	7 (7.2%)	7 (7.4%)	10 (11.4%)	10 (12.3%)	6 (11.3%)	2 (14.3%)	207	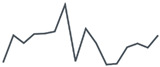
Serogroup D	3 (23.1%)	22 (18.5%)	33 (25.6%)	29 (21.2%)	36 (25.2%)	65 (41.1%)	41 (35.7%)	29 (25.7%)	35 (32.7%)	44 (36.4%)	40 (41.2%)	35 (36.8%)	27 (30.7%)	23 (28.4%)	14 (26.4%)	5 (35.7%)	481	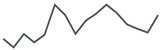
Serogroup E	1 (7.7%)	3 (2.5%)	4 (3.1%)	1 (0.7%)	4 (2.8%)	2 (1.3%)	2 (1.7%)	3 (2.7%)	1 (0.9%)	1 (0.8%)	1 (1.0%)	4 (4.2%)	9 (10.2%)	20 (24.7%)	9 (17.0%)	3 (21.4%)	68	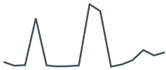
Total	13	119	129	137	143	158	115	113	107	121	97	95	88	81	53	14		

The prevalences of serogroup B, serogroup D and serogroup E showed significant differences between years (*p* = 0.000, *p* = 0.001 and *p* = 0.000 respectively). By post hoc comparison, the prevalence of serogroup B demonstrated a decreasing trend (*p* = 0.000) while those of se.

**Table 2 children-09-01403-t002:** Association between antimicrobial non-susceptibility to different serogroups of NTS.

	Serogroup B			Serogroup C1			Serogroup C2			Serogroup D			Serogroup E		
Non-Susceptibility	B	Non-B	*p*-Value	C1	Non-C1	*p*-Value	C2	Non-C2	*p*-Value	D	Non-D	*p*-Value	E	Non-E	*p*-Value
Ampicillin	66.4%	24.8%	0.000 *	15.1%	45.6%	0.000 *	29.0%	45.0%	0.000 *	20.8%	52.5%	0.000 *	60.3%	42.1%	0.004 *
Chloramphenicol	51.7%	13.4%	0.000 *	11.5%	31.9%	0.004 *	26.6%	30.6%	0.256	2.3%	42.4%	0.000 *	55.9%	28.9%	0.000 *
Ciprofloxacin	2.6%	1.6%	0.152	2.2%	2.0%	0.756	2.9%	1.9%	0.296	0.2%	2.8%	0.000 *	5.9%	1.8%	0.045 *
Ceftriaxone	9.7%	3.0%	0.000 *	4.3%	6.1%	0.571	3.4%	6.3%	0.114	1.9%	7.7%	0.000 *	7.4%	5.9%	0.596
Sulfamethoxazole-Trimethoprim	29.9%	18.7%	0.000 *	23.0%	23.6%	0.917	25.6%	23.3%	0.482	8.9%	29.9%	0.000 *	57.4%	22.0%	0.000 *

* Statistically significant (*p* < 0.05). Serogroup E was associated with significantly higher non-susceptible rate than non-E serogroup in ampicillin, chloramphenicol, ciprofloxacin and TPM/SMX. On the contrary, serogroup D was less resistant to all tested antimicrobials compared to others.

**Table 3 children-09-01403-t003:** Antibiotics non-susceptibility rate and serogroups between iNTS and non-iNTS.

	iNTS			Non-iNTS		
	2004~2011	2012~2019	*p*-Value	2004~2011	2012~2019	*p*-Value
**Antibiotics non-susceptibility**						
Ampicillin	29.1%	29.4%	1.000	39.4%	49.1%	0.000 *
Chloramphenicol	27.3%	29.4%	1.000	34.1%	24.9%	0.000 *
Ciprofloxacin	14.5%	17.6%	0.714	1.1%	1.7%	0.379
Ceftriaxone	5.5%	0.0%	1.000	6.1%	5.9%	1.000
Sulfamethoxazole-Trimethoprim	23.6%	29.4%	0.750	23.6%	23.3%	0.902
**Serogroup**						
B	41.8%	23.5%	0.253	47.6%	38.5%	0.000 *
C1	7.3%	23.5%	0.083	8.5%	8.9%	0.782
C2	29.1%	23.5%	0.764	13.4%	11.0%	0.156
D	21.8%	29.4%	0.527	28.2%	34.1%	0.015 *
E	0.0%	0.0%	1.000	2.3%	7.5%	0.000 *

* Statistically significant (*p* < 0.05).

## Data Availability

The data that support the findings of this study are available upon request from the corresponding author.

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
