# Peer review of "The Clinical Implication of Serogroup Distribution and Drug Resistance of Non-Typhoidal Salmonella in Children: A Single Center Study in Southern Taiwan during 2004–2019"

_children, 2022, doi:10.3390/children9091403_

Round 1

Reviewer 1 Report

The manuscript of M.C. Lee et al. titled "The clinical implication of serogroup distribution and drug resistance of non-typhoidal salmonella in children: a single center study in southern Taiwan during 2004-2019" is devoted to serogrouping and drug resistance of NTS and iNTS isolated from children in Southern Taiwan in 2004-2019. The manuscript is primarily well written although I recommend minor English editing since the authors are not native speakers.
Below are the minor issues and suggestions aimed at improving the manuscript:

Because I suspect the authors are not microbiologists, I would suggest capitalizing and italicizing Latin names of Salmonella throughout the text, including the title of the manuscript.

Line 38: Add full "non-typhoidal Salmonella" before (NTS).
Lines 42-43: Italicize the Latin names of the microorganisms.
Line 83: Add a reference to CSLI guidelines.
Line 96: Italicize Salmonella enterica.
Table 1: I suggest adding trend lines to the plots.
Line 155: Replace "~" with "-".

Author Response

Thank you for your comment and suggestion and sorry for my inexperience in article writing. I have revised the manuscript under your guidance. Table 1 has also been modified under your advice and reviewer 2.

Reviewer 2 Report

Dear Authors,

 The manuscript ID: children-1878772_v1 entitled „The clinical implication of serogroup distribution and drug resistance of non-typhoidal salmonella in children: a single center study in southern Taiwan during 2004-2019” written by Meng-Chien Lee, Zon-Min Lee, Yi-Chun Yeh, Hong-Ren Yu and Kuang-Che Kuo is interesting.

Salmonella enterica causes a wide range of diseases, from self-limiting gastroenteritis to invasive infections caused by non-typhoidal serovars (NTS) and typhoidal serovars, respectively. Host factors strongly influence infection outcome as malnourished, immunocompromised individuals or children can develop invasive infections from NTS. Moreover, their resistance to antimicrobial drugs is increasing. Therefore, this problem is very significant.

The whole manuscript (Introduction, Materials and Methods, Results, Discussion and Conclusions) is properly organized. Introduction contains general data on the global increase of antimicrobial resistance in NTS. The purpose of the work is concrete. The retrospective study was conducted with statistical analyses. Results were presented in the form of tables and figures and discussed. Based on them conclusions were drawn.

However, I have some suggestions and comments in order to improve paper, which are the following:

1)   In the Table 1, please add one column and one row with data: “Total”. Add the total number of isolates for each serogroup (column) and the total number of isolates for each year (row). Moreover, this table should be corrected graphically;

2)   In conclusions, please provide your own results and do not cite other authors. Please move the citations from here to the discussion;

3)   The all text - a "period" should be placed at the end of the sentence after the references, ie: “[1,2,3, etc.].” instead “.[1,2.3, etc]”

e.g.:

Line 39: annually.[1-3] – annually [1-3].

Line 43: Yersinia enterocolitica.[4] – Yersinia enterocolitica [4].

Line 44: hospitalization.[5] – hospitalization [5].

etc…..

4)   The names of the microorganisms should be written in italics:

Lines 42-43: Campylobacter (28%), Escherichia coli 0157, Shigella and Yersinia enterocolitica

Lines 80, 96, etc.: Salmonella or Salmonella enterica

5)   Other:

Line 68: Cancer – cancer

In my opinion, the obtained results are valuable and manuscript is worth publishing in „Children”.

With highest regards,

Author Response

Thank you for your comment and suggestion and sorry for my inexperience in article writing. I have revised the manuscript under your guidance. Table 1 has also been modified under your advice and by the other reviewers.